# Fermented Gamma Aminobutyric Acid Improves Sleep Behaviors in Fruit Flies and Rodent Models

**DOI:** 10.3390/ijms22073537

**Published:** 2021-03-29

**Authors:** A-Hyun Jeong, Jisu Hwang, Kyungae Jo, Singeun Kim, Yejin Ahn, Hyung Joo Suh, Hyeon-Son Choi

**Affiliations:** 1Department of Public Health Science, Korea University, Seoul 02841, Korea; ah20@korea.ac.kr (A.-H.J.); jisuuu0731@gmail.com (J.H.); jo-kyung-ea@hanmail.net (K.J.); kimsingun@gmail.com (S.K.); cassandra7@hanmail.net (Y.A.); suh1960@korea.ac.kr (H.J.S.); 2Interdisciplinary Program in Precision Public Health, Korea University, Seoul 02841, Korea; 3Department of Food Nutrition, Sangmyung University, Seoul 03016, Korea

**Keywords:** γ-aminobutyric acid, GABA receptor, sleep promotion, non-rapid eye movement, *Drosophila melanogaster*, rodents

## Abstract

The aim of this study was to investigate the effect of *Lactobacillus brevis*-fermented γ-aminobutyric acid (LB-GABA) on sleep behaviors in invertebrate and vertebrate models. In *Drosophila melanogaster*, LB-GABA-treated group showed an 8–9%-longer sleep duration than normal group did. LB-GABA-treated group also showed a 46.7% lower level of nighttime activity with a longer (11%) sleep duration under caffeine-induced arousal conditions. The LB-GABA-mediated inhibition of activity was confirmed as a reduction of total movement of flies using a video tracking system. In the pentobarbital-induced sleep test in mice, LB-GABA (100 mg/kg) shortened the time of onset of sleep by 32.2% and extended sleeping time by 59%. In addition, mRNA and protein level of GABAergic/Serotonergic neurotransmitters were upregulated following treatment with LB-GABA (2.0%). In particular, intestine- and brain-derived GABA_A_ protein levels were increased by sevenfold and fivefold, respectively. The electroencephalography (EEG) analysis in rats showed that LB-GABA significantly increased non-rapid eye movement (NREM) (53%) with the increase in theta (θ, 59%) and delta (δ, 63%) waves, leading to longer sleep time (35%), under caffeine-induced insomnia conditions. LB-GABA showed a dose-dependent agonist activity on human GABA_A_ receptor with a half-maximal effective concentration (EC_50_) of 3.44 µg/mL in human embryonic kidney 293 (HEK293) cells.

## 1. Introduction

γ-Aminobutyric acid (GABA) is an amino acid derivative synthesized in nature by γ-Aminobutyric acid (GABA) is an amino acid derivative synthesized in nature by microorganisms, plants, and animals [1]. GABA has been reported to have various physiological activities [2] in addition to playing the role of an endogenous inhibitory neurotransmitter in the central nervous system of mammals [1]. GABA has been reported to enhance the synthesis of proteins including growth hormones [2]. In addition, GABA is known to contribute to the survival and replication of cells, with antidiabetic effects [3,4]. However, the GABA level in the brain has been known to decrease with aging [5] while the use of GABA supplements has been known to improve blood pressure [6], depression [7], and immunity [8].

Recently, GABA has been used in pharmaceuticals and as an active constituent in foods with increasing demand [9]. However, GABA-containing foods cannot meet the considerable demand because they contain very small amounts of GABA. One of the most common ways to produce GABA is fermentation by yeast, fungi, and bacteria [10,11]. In particular, lactic acid bacteria (LAB) have been widely used as GABA-producing microorganisms in the food industry because of their merits such as a generally recognized as safe (GRAS) status and health-oriented properties [10].

Sleep is one of the most important physiological processes for a healthy life [12]. Sleep restores the body systems that have been weakened during daily life activities and facilitation the maintenance of endocrine, cognitive, and immune functions [13]. Moderate sleep enhances the learning ability and memory of the brain [12] and supports the normal growth and development of the body with hormonal balance [14]. Recently, lack of the sleep has been identified as a common sleep disorder in modern populations and is becoming an increasing health issue worldwide. Approximately one-third of the general population is currently experiencing difficulty in falling or staying asleep [15].

Constant sleep deficiency has adverse effects and increases various health risks such as heart disease, stroke, obesity, and hypertension [16], and is strongly linked to mental health such as depression [17]. In addition, sleep deprivation can be detrimental to a successful occupational life as it may result in loss of concentration at work with reduced productivity and even automobile accidents [16]. Insomnia is usually treated pharmacologically, and benzodiazepines (BZDs) and zolpidem are the most widely prescribed medications as hypnotics for the induction of sleep [17].

However, prolonged use of these drugs induces side effects such as cognitive decline, drug dependence or addiction, daytime lethargy, and memory loss [17]. Thus, nonpharmacological treatments have been suggested as alternatives with milder side effects. Herbal supplements and food materials have been known to promote sleep with improvement of sleep behavior by regulating central GABAergic transmission [18]. A recent study showed that sleep latency and duration can be modified by nutritional diets [19], indicating that nutritional recipes can improve sleep behaviors. In contrast, caffeine, a major component of coffee, disturbs sleep [20].

The connection between GABA and sleep has been reported by several studies [7,21,22]. A previous study showed that a mixture of GABA and theanine promotes sleep in rodent models. However, the observed actions were not GABA-dependent [21]. An in-human study showed that GABA favorably affected sleep latency and non-rapid eye movement (NREM) sleep time, but there is no information of total sleep time with the deteriorated sleep parameters of placebo group [22]. Thus, more systematic studies are necessary to clarify and confirm the relationship between GABA and sleep behaviors as well as the underlying mechanistic actions in various experimental models.

In this study, the effect of Lactobacillus-fermented GABA (LB-GABA) was systematically examined on the sleep behaviors of fruit flies, mice, and rats, including the analysis under caffeine-induced insomnia conditions. In addition, the agonist activity of LB-GABA on the human GABA_A_ receptor, a major sleep-promoting neurotransmitter receptor, was analyzed in a human embryonic kidney 239 (HEK293) cell culture system.

## 2. Results

### 2.1. Analysis of LB-GABA

The GABA content of the LB-GABA samples was analyzed using an HPLC system and was detected as a major peak (Figure 1). The elution time for GABA was 23.95 min (Figure 1B), which was the same as that of the synthetic GABA (Figure 1C). Thus, the LB-GABA sample was confirmed to be GABA by comparing it to the standard synthetic sample.

### 2.2. Effects of LB-GABA on Locomotor Activity of Fruit Flies

The actogram showed that BZD and LB-GABA reduced nighttime and daytime locomotor activity in the LAM system (Figure 2A). In particular, high-dose LB-GABA decreased the nighttime locomotor activity of fruit flies by 37.2% compared to that of the NOR group (Figure 2B). The inhibitory effect of LB-GABA on locomotor activity was comparable to that of alprazolam, a sleep-inducing BZD agent. Sleep bout, which is the frequency of arousal during sleep, showed a decreasing trend in the BZD and high-dose LB-GABA groups, compared with NOR group, but the difference was not significant (Figure 2C). The nighttime sleep duration was significantly longer in the BZD and LB-GABA groups than it was in the NOR group. Treatment with BZD (0.1%) and LB-GABA (2.0%) increased the sleep time by 8–9% compared with that of the normal group (Figure 2D). This result showed that LB-GABA increased the nighttime sleep duration compared with the NOR group.

In the daytime, BZD and LB-GABA also inhibited the locomotor activity, but the reduction induced by LB-GABA was not significant. Only BZD treatment significantly decreased the daytime locomotor activity and sleep bouts (Figure 2E,F). Sleep duration was not significantly different among the groups except for that of the BZD group. (Figure 2G). This result indicated that LB-GABA did not significantly promote daytime sleep.

### 2.3. Effects of LB-GABA on Locomotor Activity in Caffeine-Induced Awake Model of Fruit Flies

The caffeine-treated group showed a significant increase (59.2%) in nighttime activity compared with that of the NOR group (Figure 3A), indicating that caffeine induced a wakeful state in the flies. Caffeine-induced increased activity was decreased in the LB-GABA-treated groups in a dose-dependent manner. High-dose LB-GABA (2.0%) reduced the nighttime activity by 45.8% compared with that of the caffeine only treated group. This result showed that LB-GABA effectively suppressed the caffeine-induced increase of nighttime activity.

Caffeine increased sleep bouts, but the effect was not statistically significant, whereas additional BZD treatment failed to alleviate the caffeine-induced increase of sleep bouts. LB-GABA/caffeine cotreatment decreased the sleep bout, but the effect was not statistically significant (Figure 3B). The sleep duration of caffeine-treated flies was significantly lower than that of the NOR flies, but BZD and LB-GABA recovered the sleep time, and high-dose LB-GABA increased the sleep duration by approximately 11% compared with that of the caffeine-treated group (Figure 3C). This result showed that LB-GABA had a sleep-recovering effect even under caffeine-induced insomnia conditions.

### 2.4. Effects of LB-GABA on Movement of Fruit Flies

The distance moved by the flies of NOR group was greatly increased by caffeine treatment (48.2%, Figure 4A). The caffeine-induced increase in the moved distance was decreased by 50.7% and 55.3% with BZD and LB-GABA treatment, respectively. (Figure 4A). Treatment with LB-GABA alone decreased (23%) the movement of the flies compared with that of the NOR flies. Caffeine treatment effectively increased the movement velocity and time, but this effect was significantly suppressed in the BZD and LB-GABA groups (Figure 4B,C). LB-GABA inhibited the caffeine-induced increase of movement velocity and time by 45% and 38%, respectively compared with caffeine treatment alone (Figure 4B,C). The time during which the flies were immobile showed an opposite trend to their movement time among the groups (Figure 4D). This result showed that LB-GABA had a calming effect on the movement of the flies.

### 2.5. Effect of LB-GABA on mRNA Expression of Neurotransmitter Receptors in Fruit Flies

BZD and high-dose LB-GABA (2.0%) treatment increased the mRNA expression of Rdl, a GABA_A_ receptor, by 81% compared with that in the NOR group in fruit flies (Figure 5A). GABA_B_-R1, a GABA_B_ receptor, and 5HT-1A, a serotonin receptor, were also significantly increased with LB-GABA and BZD treatment in a dose-dependent manner (Figure 5B,C). High-dose LB-GABA (2.0%) increased GABA_B_-R1 and 5HT-1A mRNA expression by 79% and 57%, respectively, compared with that of the NOR group (Figure 5B,C). This result showed that LB-GABA promoted sleep in flies by upregulating GABAergic and serotonergic receptors.

### 2.6. Effects of LB-GABA on Pentobarbital-Induced Sleeping Behavior of Mice

The effects of LB-GABA on sleep latency and duration were determined using hypnotic doses (42 mg/kg, intraperitoneally (i.p.)). BZD and LB-GABA (GH, 100 mg/kg) treatment effectively shortened sleep latency time by 43.6% and 32.2%, respectively, compared with the NOR group (Figure 6A). This result showed that LB-GABA and BZD accelerated the onset of pentobarbital-induced sleep (Figure 6A) and increased the total sleep duration by 79% and 52%, respectively, compared with that of the NOR group (Figure 6B). This result indicated that LB-GABA effectively promoted sleep by shortening sleep latency and extending sleep duration under pentobarbital-induced hypnotic conditions.

### 2.7. Effects of LB-GABA on mRNA Expression and Protein Abundance of Neurotransmitter Receptors in Mice

GABA_A_ receptor mRNA level from small intestine and brain of mice was increased by 29% and 40%, respectively, compared with the NOR group in high-dose treatment of LB-GABA (100 mg/kg) (Figure 7A). The mRNA levels of GABA_B_-R1, and 5HT-1A receptors were also increased by 18% and 17%, respectively, compared with the NOR group in high-dose of LB-GABA, but 5HT-1A mRNA expression change was not statistically significant (Figure 7A). Low-dose of LB-GABA (60 mg/kg) did not show significant increase in mRNA expression of receptors, except for brain-derived GABA_A_ expression showing an 18%-increase compared with the NOR group (Figure 7A). For protein levels, high-dose LB-GABA significantly increased protein expressions of receptors (Figure 7B). The protein abundance of GABA_A_ (small intestine), GABA_A_ (brain), GABA_B_-R1α, GABA_B_-R1β, and 5HT-1A protein was increased by 7-fold, 5-fold, 1.5-fold, 2-fold, and 1.5-fold, respectively, compared with the NOR group (Figure 7C). This result showed that LB-GABA promoted sleep in mice by upregulating GABAergic and serotonergic receptors.

### 2.8. Effects of LB-GABA on Sleep Pattern of Rats

LB-GABA promoted sleep by increasing sleep time and decreasing arousal time in the normal rat model (Figure 8A,B). The duration of REM and NREM sleep was significantly increased in the BZD (alprazolam 300 μg/kg) and LB-GABA (200 mg/kg) treatment groups (Figure 8C,D). The high-dose LB-GABA (GH) group showed an 18%- and 11%-increase in REM and NREM, respectively, compared with that of the NOR group. The brain waves of the LB-GABA (200 mg/kg)-treated group showed a significant increase in the θ wave (Figure 8E), but the δ wave, which indicates deep sleep, was not significantly changed by LB-GABA treatment (Figure 8F). Therefore, the LB-GABA-induced increase of NREM was mediated by an increase in the θ wave rather than the δ wave (Figure 8E,F). In the caffeine-induced insomnia model, LB-GABA showed a more obvious sleep-promoting effect than that in the normal condition. LB-GABA (200 mg/kg) increased sleep duration by 35% and decreased arousal time by 32%, compared with the caffeine only treated group (Figure 9A,B). The LB-GABA-mediated increase in sleep time was caused by an increase in the NREM rather than REM duration, in contrast to the effects observed in the normal condition (Figure 9C,D). NREM duration in the LB-GABA (200 mg/kg)-treated group increased by 53% compared with that in the caffeine only treated group (Figure 9D). BZD also showed a 63%-increase in NREM duration compared to the that of the caffeine only treated group (Figure 9D). LB-GABA-induced increase of NREM was mediated by an increase in both θ and δ waves (59% and 63%, respectively, Figure 9E,F).

### 2.9. Agonist Effects of LB-GABA on Human GABA_A_ Receptor

LB-GABA induced a dose-dependent increase in agonist activity at up to 30 μg/mL, indicating that, at EC100 concentrations and above (100 to 1000 μg/mL), the cell culture system was saturated and reached a plateau (Figure 10A). The EC_50_ of LB-GABA was determined to be 3.44 μg/mL (Figure 10A). This result showed that LB-GABA exhibited an agonist effect on the human GABA_A_ receptor. The agonistic activity of synthetic GABA was analyzed to identify the maximal effective concentration (EC_100_) on the human GABA_A_ receptor (Figure 10B). Synthetic GABA increased the current response in a dose-dependent manner within the range of 0.1 to 30 M, with an EC100 of 30 μM (Figure 10B). In addition, the half-maximal effective concentration (EC_50_) of synthetic GABA was calculated to be 7.7 μM (Figure 10B).

## 3. Discussion

This study showed the effect of LB-GABA on sleep behaviors and time in *D. melanogaster*, mice (ICR), and rats (SD) models, including the analysis of brain waves and agonistic activity on the human GABA_A_ receptor. LB-GABA improved the nighttime sleep behaviors of flies and decreased the locomotor activity, which was associated with the increased nighttime sleep duration. Such an LB-GABA-mediated calming effect was also observed in the DAM system and open field test using caffeine as a stimulant. Caffeine is a psychoactive stimulant used to maintain alertness [23,24]. It is a major component of coffee, which is one of the most widely consumed beverages [23]. The prevalence of sleep disorders in the population is partly related to the increase in coffee consumption [24]. Our data consistently showed that LB-GABA suppressed the caffeine-induced disorders of sleep behaviors by reducing the locomotor activity of fruit flies, while recovering the sleep time. These results indicated that LB-GABA improved sleep behaviors under both normal and insomnia conditions. In addition, LB-GABA enhanced the sleep-inducing activity of pentobarbital by increasing the sleep onset ratio, through shortening of the sleep latency and prolongation of the sleep time. This result indicated that cotreatment with LB-GABA may reduce the required dose of pentobarbital by enhancing its actions.

Sleep architecture is composed of two repetitive periods of REM and NREM sleep [25]. The NREM pattern was characterized by a slow wave, which was observed as a slow cortical oscillation in the EEG analysis. This is known to be the stage for relaxed and deep sleep [26]. REM is the desynchronized type of brain wave that occurs during dreaming in which fast rhythms are formed through cortical and hippocampal interactions [27]. Sleep quality is dependent on the relative level of these two patterns during sleep [25]. Our data showed that LB-GABA increased the sleep duration by increasing both NREM and REM under normal conditions. This result shows that LB-GABA increased sleep duration by enhancing the overall sleep architecture. However, LB-GABA was shown to selectively increase NREM rather than REM sleep in the caffeine-induced insomnia model, indicating that LB-GABA contributed more favorably to sleep promotion under insomnia conditions. This result correlated with those of an investigation of a GABA/theanine mixture that showed an increase in NREM [21]. The effect of GABA on NREM was also partially verified in a previous human study, although the effect of GABA on total sleep time was not mentioned [22]. The effect of LB-GABA on the brain wave of NREM in the current study was different under normal and caffeine-induced conditions. LB-GABA mainly regulated the θ wave to increase NREM under normal conditions, while the increase in the δ wave was more obvious under caffeine-induced insomnia conditions. This result indicated that LB-GABA ameliorated the caffeine-induced sleep disturbance with enhancement of sleep quality.

The interaction of GABA and GABA receptors has been known to dominate sleep behaviors [28]. GABA receptors are generally divided into two types, GABA_A_ and GABA_B_. The responsiveness of GABA_A_ to GABA has been known to promote sleep onset in the early stage of sleep, whereas GABA_B_ is known to contribute to maintaining sleep in the mid-term and terminal stages of sleep [28]. LB-GABA upregulated the mRNA levels of these GABA receptors, indicating that its sleep-promoting effect was mediated via GABAergic interactions.

LB-GABA exhibited a dose-dependent agonist activity on human GABA_A_ receptors. The EC_50_ values of LB-GABA and the synthetic standard GABA were calculated to be 3.4 μg/mL and 7.7 μM, respectively. Assuming LB-GABA were a single compound (molecular weight [MW], 103.12), its EC_50_ would be 33.3 μM. Thus, considering that LA-GABA contains 20% GABA, it showed a similar or stronger agonistic activity than that of synthetic GABA on the human GABA_A_ receptor. Karim et al. [29] reported that the EC_50_ of GABA on human recombinant GABA_A_ receptors expressed in *Xenopus* oocytes was approximately 107 μM, which indicated the response was weaker than that observed in our study based on the EC_50_ of LB-GABA. This discrepancy may be attributable to the difference in cell system or GABA source used. The agonistic activity of GABA on GABA_A_ receptors has been reported to depend on the subunit, and α and β subunits are generally known to contribute to the EC_50_ of GABA on GABA_A_ receptor [30]. Moreover, our data showed that the 5HT-1A serotonin receptor was also upregulated by LB-GABA. The interaction between serotonin and 5HT-1A has been known to cause divergent results on sleep behaviors [31], with increased deep sleep or wakening. The duplicity of the 5HT-1A receptor response is attributable to its position in the central nervous system (CNS) [31]. The LB-GABA-mediated increase in 5HT-1A receptor expression likely positively contributes to its sleep promoting effects. These results indicated that LB-GABA-mediated sleep promotion may be mediated by multiple mechanisms.

GABA, an important molecule for sleep regulation, reduces neuronal activity by suppressing excitatory neurotransmitters such as glutamate, thereby promoting sleep [1]. Thus, the GABA level is an important factor for sleep regulation; people with insomnia have lower GABA levels than those with normal sleep [32]. The GABA-mediated sleep-promoting effect is known to be due to the GABAergic action involving the interaction with the GABA receptor and its response. However, it is questionable whether GABA directly acts on CNS-derived receptors because there is some controversy about its ability to cross the blood–brain-barrier (BBB) [33]. Nevertheless, many studies including the present work have shown the profound effects of GABA [7,21,22]. Although GABA poorly penetrates the BBB, GABA has been known to affect brain physiology through various routes [34]. In particular, GABA receptors are also expressed in the enteric system (gut) [34], indicating that GABA may indirectly affect the brain via the gut–brain axis [35]. GABA-mediated reciprocal influence between gut and brain may be supported by our data, showing that intestine-derived GABA_A_ receptor was increased in genetic and protein expression with LB-GABA treatment. In addition, GABA has been known to contribute to the production of melatonin, a sleep-related hormone, by promoting the synthesis of *N*-acetylserotonin, a precursor of melatonin, from serotonin [36]. However, there are still some limitations on GABA-mediated sleep promotion from this work. Firstly, what routes does GABA take to induce sleep promotion, via BBB or/and gut–brain axis? Secondly, besides GABAergic mechanism, how involved is the other pathway such as serotonergic action in GABA-mediated sleep promotion? The further analyses on GABA level in brain/blood and bidirectional signaling of gut–brain including serotonergic pathway would provide a valuable information to clarify above questions on how GABA promotes sleep in future study.

Many studies have reported the sleep-promoting effects of natural products, such as those reporting that extracts of *Ecklonia cava* Kjellman, Passionflower (*Passiflora incarnata*), valerian/hop, *Polygonatum sibiricum* have been known to increase sleep duration in mice or rats [37,38,39,40]. These herbal extracts have been known to exhibit sleep promoting effects via GABAergic actions, similar to those observed in this study. These natural products might also contain GABA, which could, at least party, contribute to the sleep-promoting effects. *Nelumbo nucifera* seed extract, which showed sleep-promoting effects, was found to contain GABA as an active component [41]. Thus, for GABAergic activity-targeted promotion of sleep, fermentation-derived GABA is thought to be one of the most excellent options.

## 4. Materials and Methods

### 4.1. Preparation and Analysis of LB-GABA

The LB-GABA sample was produced using *Lactobacillus brevis* fermentation from monosodium L-glutamate (Medience Co., LTD, Chuncheon, Korea). The LB-GABA is composed of ~20% GABA (*w*/*w*) and ~80% modified food starch (AMOREPACIFIC Co., Seoul, Korea). GABA from LB-GABA was analyzed using Alliance HPLC system (Waters, Milford, MA, USA) equipped with an auto-sampler, quaternary pump, and a UV detector. Dansyl chloride derivatives GABA was detected using the Mightysil RP-18 column MG (4.6 × 250 nm) and UV detector (254 nm). The mobile phase A was a mixture of tetrahydrofuran, methanol and 50 mM sodium acetate (pH 6.2), (5:75:420, *v*/*v*). The mobile phase B was a methanol. The flow rate was 1.0 mL/min. The gradient of mobile phase was performed as follows: 20% B for 0–6 min, 20–80% B for 6–20 min, 80–100% B for 20–23 min, 100–20% B for 23–24 min, 20% B for 24–35 min.

### 4.2. Fly Maintenance

The *D. melanogaster* Canton-S strain of flies (wild-type) were obtained from the Bloomington *Drosophila* Stock Center (Indiana University, Bloomington, IN, USA). The medium composed of sucrose, cornmeal, dried yeast, agar, propionic acid, and *p*-hydroxybenzoic acid methyl ester solution was freely supplied as food for the flies, which were maintained at 25 °C with a 60% relative humidity under a 12 h light/dark cycle. The LB-GABA samples were added to the above medium at the indicated concentrations. Each 2-day-old male fly was anesthetized using CO_2_ for the subsequent analysis.

### 4.3. Determination of Sleep Behavior

The fruit flies were treated with LB-GABA (0.5, 1.0, and 2.0%) and alprazolam 0.1% (BZD) as the test and positive control samples, respectively, to determine their sleep behavior as previously described [41]. Flies were adapted for 3 days in the tubes at 25°C, and their activities were recorded every 1 min for 3–5 days in the dark. Caffeine (0.1%) was used to induce the arousal state. Data were analyzed using the DAM management software (TriKinetics, Waltham, MA, USA) while the interruptions were recorded using an infrared detector and visualized using the Actogram J software. The following sleep parameters, total activity, sleep bout, and sleep time were analyzed using the R statistical software.

The open field moving test of the flies was performed using a video-tracking system as reported in a previous study [42] with some modifications. Flies treated with or without caffeine were further treated with BZD or LB-GABA and after adaptation to the chamber for 1 min, the activity of 10 flies per group was visually recorded for 5 min, 1 h before subjective nighttime. The open field chamber was a circular arena (8 mm × 0.1 mm, diameter and height) with a white background. The mobility of the flies was monitored using the Noldus EthoVision-XT system (Noldus Information Technology, Wageningen, The Netherlands).

### 4.4. Rodent Maintenance

Male ICR mice (6-week-old, weighing 27–32 g) and Sprague-Dawley rats (8-week-old, weighing 250–270 g) were obtained from Orient Bio Inc. (Seongnam, Korea). All animals were housed in cages maintained at 24 °C with a relative humidity of 55% under a 12-h light/dark cycle. Food and water were freely provided. The animals were allowed to acclimate to the vivarium for at least 1 week before the pentobarbital-induced sleep test or electroencephalographic (EEG) analysis. All the animal experiments were approved by the Korea University Animal Care Committee (KUIACUC-2020-0006, Seoul, Korea).

### 4.5. Pentobarbital-Induced Sleep Determination

The pentobarbital-induced sleep test was performed between 1:00 and 5:00 p.m., and the ICR mice were starved for 1 day prior to the test. All the samples were resuspended in physiological saline (0.9% sodium chloride [NaCl]) for oral administration. The test groups were the normal, BZD (alprazolam, 0.2 mg/kg), and LB-GABA (60 and 100 mg/kg) and 45 min after each group was treated, pentobarbital (42 mg/kg) was injected into the left side of the abdomen.

The mice were placed in individual cages and monitored to measure sleep latency and duration. Sleep was determined as the time that elapsed between righting reflex loss and recovery. Sleep latency was defined as the time between the pentobarbital injection and sleep onset. Mice that failed to sleep within 10 min after the pentobarbital injection were removed from the analysis [43].

### 4.6. EEG Analysis

The surgery for EEG analysis was performed as previously described [39]. After the surgery, all the animals were treated with an antibiotic and placed in individually cages with water and food for 7 days. The rats were randomly divided into the normal control (NOR), alprazolam-treated (BDZ), and low-and high-dose LB-GABA-treated (GL and GH) groups. The analysis was performed between 10 a.m. and 5 p.m. for 9 days. LB-GABA was administered orally 1 h before the analysis and the EEG signals were amplified, filtered (0.5–30.0 Hz), and recorded using Iox2 software (version 2.8.0.13, Emka Technologies, Paris, France).

The EEG spectra were analyzed in 1 Hz frequency bins with the standard frequency bands set as following: γ: 30–60 Hz, β: 12–30 Hz, α: 9–12 Hz, θ: 4–9 Hz, and δ: 0.5–4 Hz. The sleep or wake time wave pattern was quantified each time in the range of 0–30 Hz for 10 s intervals using fast Fourier transform (FFT) with the ecgAUTO3 software (version. 3.3.0.20, Emka Technologies). Caffeine (40 mg/kg) was used to induce a wakeful state before the experiments.

### 4.7. mRNA Expression of Neurotransmitter Receptors

Flies were treated with LB-GABA (0.5, 1.0, and 2.0%) for 14 days and mice were administered with LB-GABA (60 mg/kg and 100 mg/kg) during maintenance of 4 weeks. Total RNA was extracted from the heads of the flies or small intestine/brain of mice using TRIzol^®^ reagent (Invitrogen, CA, USA). mRNA expression of neurotransmitters was analyzed using the Power TaqMan PCR Master Mix kit (Applied Biosystems, Foster City, CA, USA) and the StepOne plus software v 2.0 (Applied Biosystems, Foster City, CA, USA) as previously described [40]. Primers for analyzing the genes of interest of flies and mice were as follows: RpL32 (NM_001144655.3), GABAA-R Rdl (NM_001274688.1), GABAB-R1 (NM_001259104.1), and 5-hydroxytryptamine 1A (5HT-1A, NM_166322.2) for flies; GABAA (NM_008076.3), GABAB-R1 (NM_019439.3), 5HT-1A (NM_008308.4), GAPDH (NM_001289726.1) for mice.

### 4.8. Protein Abundance of Neurotransmitter Receptors

Protein (50 mg) obtained from mice brain or intestine was isolated on 10% Tris-Glycine polyacrylamide gels (Bio-rad Laboratories, Inc., Hercules, CA, USA) and transferred to Immobilon-P membranes (Millipore Corporation, Bedform, MA, USA). The membrane was incubated overnight at 4 ℃ with following antibodies: anti-GABAA-a2 (Abcam, #ab72445), anti-GABAB-R1 (Cell signaling, #3835), anti-5HT1A (Abcam, #ab85615), or GAPDH (Cell signaling, #5174). The membrane-bound antibodies were applied to secondary IgG antibody (Cell signaling, #5127) and signals were visualized using ECL Prime (GE Healthcare Life Sciences, Issaquah, WA, USA) on a FluorChem E Imaging System (Protein Simple, San Jose, CA, USA). Protein signals were quantified using FluorChem E Image Quant software with normalization by GAPDH as an internal control.

### 4.9. HEK293 Cell Culture

HEK293 cells were cultured in Dulbecco’s modified Eagle’s medium (DMEM) containing 10% fetal ovine serum (FBS) and 1% penicillin-streptomycin (PS). HEK293 cells were grown to approximately 70% confluence in dish (60 × 15 mm) for agonist test. Cells were maintained at 37 °C, 5% of CO_2_, and 95% humidity.

### 4.10. Human GABA_A_ (hGABA_A_) IonFlux HT Agonist Assay

The agonist activity of LB-GABA (20% GABA, *w/w*) on the human GABA_A_ (hGABA_A_ α1β2γ2) receptor was evaluated using the IonFlux HT electrophysiological platform (Eurofins, St Charles, MO, USA). Peak inward currents in response to the addition of LB-GABA or standard GABA were measured in HEK293 cells expressing hGABA_A_ α1β2γ2. The external recording solution consisted of NaCl (137 mM), potassium chloride (KCl, 4 mM), magnesium chloride (MgCl_2_,1 mM), calcium chloride (CaCl_2_,1.8 mM), 4-(2-hydroxyethyl)-1-piperazineethanesulfonic acid (HEPES, 10 mM), and glucose (10 mM).

The internal recording solution consisted of potassium fluoride (KF, 70 mM), KCl (60 mM), NaCl (15 mM), HEPES (5 mM), ethylene glycol tetra acetic acid (EGTA, 5 mM), and MgATP (4 mM). LB-GABA or the GABA standard was applied for 2 s followed by a 60-s wash. The process was repeated with a maximum of seven increasing concentrations of the test samples (1, 3, 10, 30, 100, 300, and 1000 μg/mL) per well.

The data were normalized to the baseline peak current induced by the addition of the EC_100_ of GABA. The normalized peak current was calculated using the following formula: normalized peak current = (Inward current ^(LB-GABA)^/Inward current ^(GABA)^). Where inward current ^(LB-GABA)^ and inward current ^(EC100 GABA)^ are the peak and baseline peak currents induced by adding LB-GABA and the EC_100_ of GABA, respectively. All the data were first exported into a Microsoft Excel compatible data file and then analyzed using GraphPad Prism software.

### 4.11. Statistical Analyses

All statistical analyses were performed using the Statistical Package for the Social Sciences (SPSS) version 12.0 (SPSS Inc., Chicago, IL, USA). Differences between groups were evaluated using a one-way analysis of variance (ANOVA) and Tukey’s multiple comparison test. All the data are presented as means ± standard deviation (SD). Different symbols indicate significant differences at * *p* < 0.05, ** *p* < 0.01, and *** *p* < 0.001 vs. normal group. Additionally, # *p* < 0.05, ## *p* < 0.01, and ### *p* < 0.001 vs. control group.

## 5. Conclusions

In conclusion, LB-GABA effectively improved sleep behaviors in fruit fly, mouse, and rat models likely via GABAergic actions. LB-GABA showed more obvious sleep-promoting effect with an improvement of sleep quality in the caffeine-induced insomnia model.

## Figures and Tables

**Figure 1 ijms-22-03537-f001:**
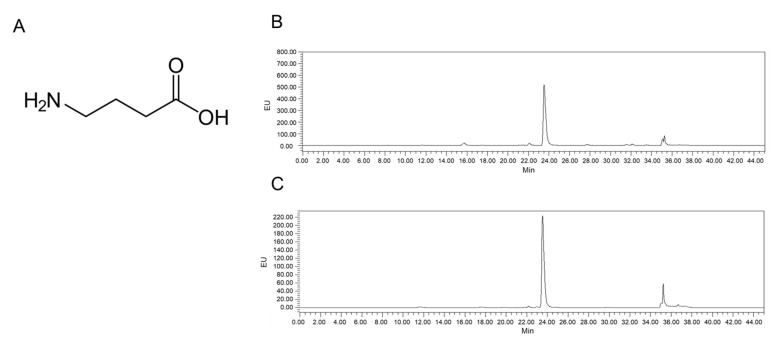
(**A**) Chemical structure and (**B**) high-performance liquid chromatography (HPLC) chromatograms of *Lactobacillus*-fermented γ-aminobutyric acid (LB-GABA), and (**C**) GABA standard.

**Figure 2 ijms-22-03537-f002:**
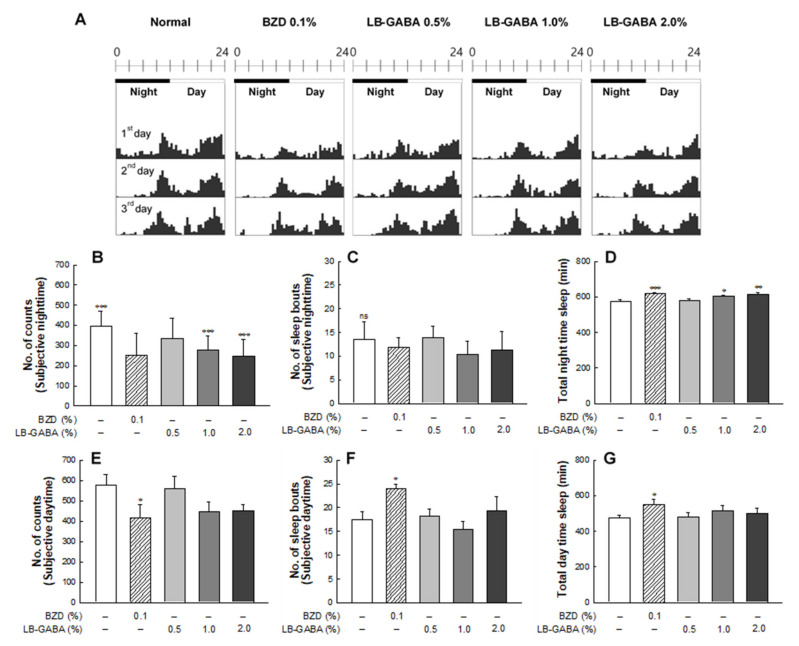
Effects of *Lactobacillus*-fermented γ-aminobutyric acid (LB-GABA) on Locomotor Activity in Fruit Flies. Constant nighttime and daytime behavioral analysis was conducted for 3 days. (**A**) Locomotor activity of control, alprazolam-treated (benzodiazepine, BZD), and LB-GABA-treated flies (*n* = 90/group) displayed in representative actograms. Activity detected per 1 min using a sensor was expressed as numbers, and then activity count for 30 min was combined and over 3 days. Actograms of each groups showed daytime (white) and nighttime (black) readings (h). (**B**) Subjective nighttime activity. (**C**) Nighttime sleep bouts. (**D**) Total nighttime sleep duration. (**E**) Daytime activity. (**F**) Daytime sleep bouts. (**G**) Total daytime sleep duration. Values are means ± standard deviation (SD); * *p* < 0.05, ** *p* < 0.01, and *** *p* < 0.001 versus normal control group (NOR) according to Tukey’s multiple comparison test; ns, not significant.

**Figure 3 ijms-22-03537-f003:**
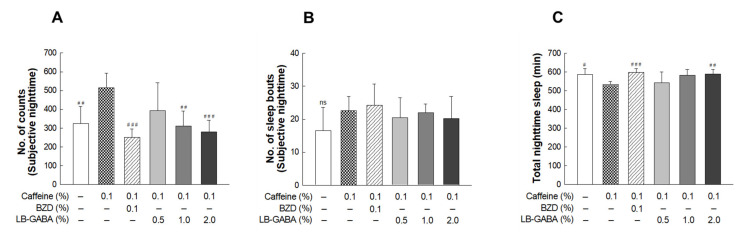
Effects of *Lactobacillus*-fermented γ-aminobutyric acid (LB-GABA) on Caffeine-induced Arousal in Fruit Flies. Locomotor activity of files was monitored for 3 days in constant darkness. (**A**) Subjective nighttime activity, (**B**) Number of sleep episodes. (**C**) Subjective nighttime sleep duration, comparing control (sucrose–agar medium), 0.1% alprazolam (benzodiazepine, BZD)-treated, and LB-GABA-treated (0.5, 1.0, and 2.0%) groups with 0.1% caffeine-treated group using *Drosophila* activity monitoring (DAM) system. Values are means ± standard deviation (SD); # *p* < 0.05, ## *p* < 0.01, and ### *p* < 0.001 versus caffeine group according to Tukey’s multiple comparison test. ns, not significant.

**Figure 4 ijms-22-03537-f004:**
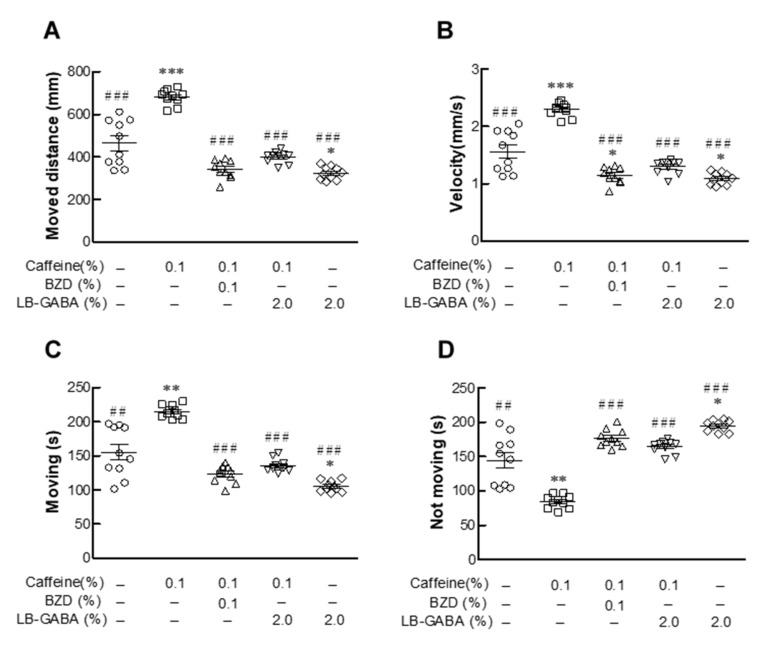
Effects of *Lactobacillus*-fermented γ-aminobutyric acid (LB-GABA) on movement of fruit files. Movement of flies was determined using a white board every 5 min for 1 h before subjective nighttime. (**A**) Moving distance (mm), (**B**) velocity (mm/s), (**C**) movement time of flies from center point, and (**D**) immobility time of flies from center point of control (sucrose–agar medium), 0.1% alprazolam (benzodiazepine BZD)-treated, and LB-GABA-treated (2.0%) groups compared with 0.1% caffeine-treated group using Noldus EthoVision-XT system. Values are means ± standard deviation (SD); * *p* < 0.05, ** *p* < 0.01, and *** *p* < 0.001 versus normal group(no treatment); ## *p*< 0.01 and ### *p* < 0.001 versus caffeine-only group according to Tukey’s multiple comparison test.

**Figure 5 ijms-22-03537-f005:**
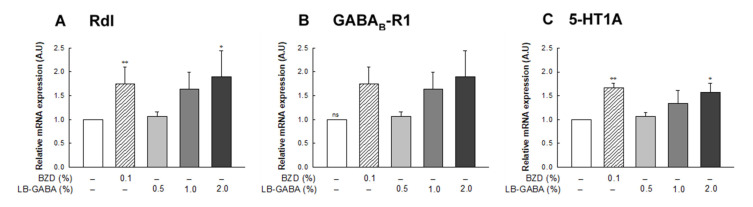
Effects of *Lactobacillus*-fermented γ-aminobutyric acid (LB-GABA) on mRNA expression of GABAergic and serotonergic receptors in fruit flies. Total RNA was extracted from heads of flies maintained on 12-h light/dark cycle for 2 weeks. Target mRNA was analyzed via quantitative real-time PCR. (**A**) resistance to dieldrin (Rdl), (**B**) GABA_B_-R1, (**C**) 5-hydroxytryptamine 1A (5-HT1A). Values are the means± standard deviation (SD) from 150 fruit flies per group; * *p* < 0.05, ** *p* < 0.01 versus normal control group according to Tukey’s multiple comparison test. A.U: arbitrary unit.

**Figure 6 ijms-22-03537-f006:**
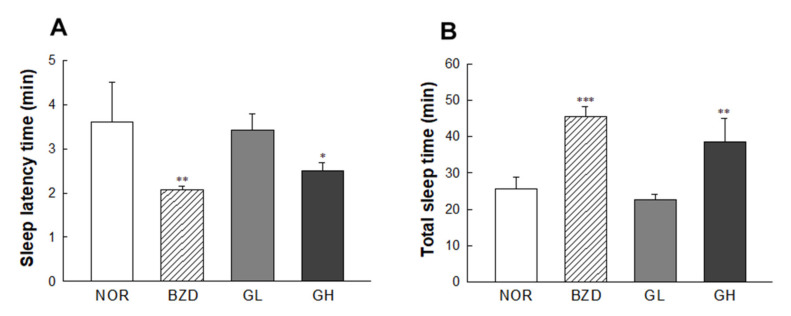
Effects of *Lactobacillus*-fermented γ-aminobutyric acid (LB-GABA) on (**A**) sleep latency time and (**B**) total sleep time in mice administered hypnotic dose of pentobarbital (42 mg/kg, intraperitoneally (i.p.)). Normal control (NOR): 0.9% sodium chloride (NaCl, physiological saline) group, BZD: benzodiazepine (alprazolam, 200 μg/kg) treatment (positive control), GL and GH: low- and high-dose LB-GABA (60 and 100 mg/kg), respectively. Values are means ± standard deviation (SD); * *p* < 0.05, ** *p* < 0.01, and *** *p* < 0.001 versus normal control group according to Tukey’s multiple comparison test.

**Figure 7 ijms-22-03537-f007:**
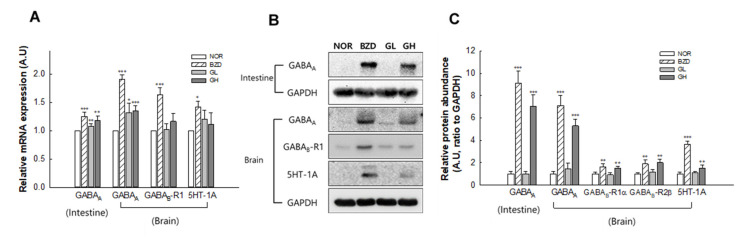
Effect of *Lactobacillus*-fermented γ-aminobutyric acid (LB-GABA) on mRNA and protein expressions of GABAergic and serotonergic receptors in mice. Target mRNAs and proteins from mice were analyzed via quantitative real-time PCR and Western blot, respectively. (**A**) mRNA expression of GABA_A_(intestine), GABA_A_(brain), GABA_B_-R1 and 5HT-1A receptors. (**B**) Protein abundances and (**C**) relative quantification of GABA_A_ (intestine), GABA_A_(brain), GABA_B_-R1α/β, and 5HT-1A receptors. NOR: normal control (0.9% sodium chloride [NaCl]) group, BZD: benzodiazepine (alprazolam, 200 μg/kg)-treated group (positive control), GL and GH: low- and high-dose LB-GABA (60 and 100 mg/kg), respectively. Values are the means ± standard deviation (SD); * *p* < 0.05, ** *p* < 0.01, and *** *p* < 0.001 versus normal control group according to Tukey’s multiple comparison test. A.U: arbitrary unit.

**Figure 8 ijms-22-03537-f008:**
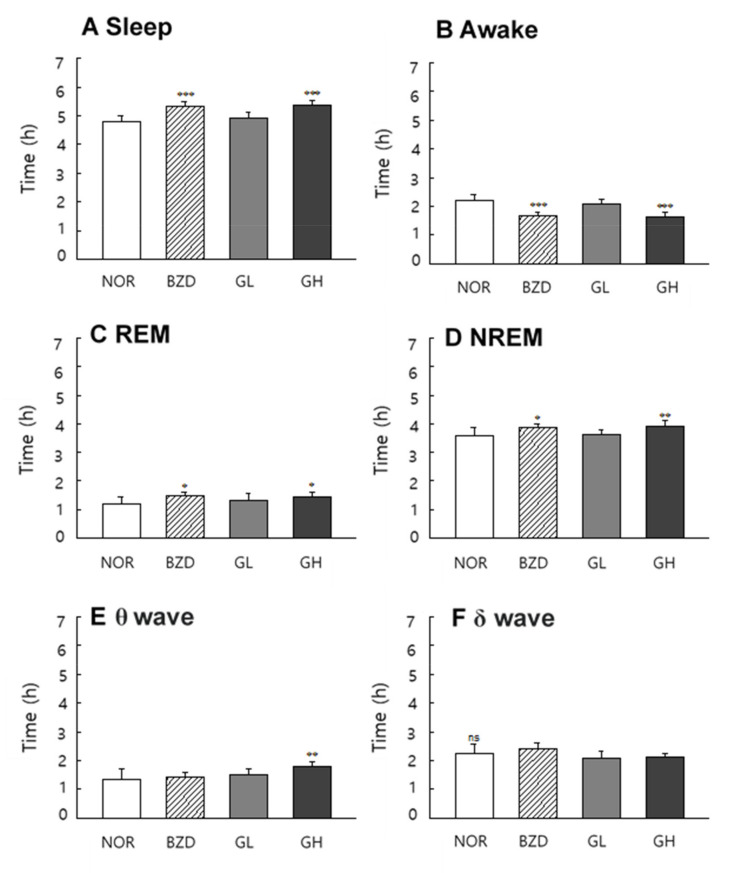
Effects of *Lactobacillus*-fermented γ-aminobutyric acid (LB-GABA) on electrophysiologic pattern in rats. LB-GABA was administered orally 1 h before experiments, and electroencephalography (EEG) analyses were conducted for 9 days. (**A**) sleep time, (**B**) awake time, (**C**) duration of REM (rapid eye movement), (**D**) duration of NREM (non-rapid eye movement), (**E**) θ wave of NERM, and (**F**) δ wave of NREM.NOR: normal control (0.9% sodium chloride (NaCl) group, BZD: benzodiazepine (alprazolam, 300 μg/kg) treatment (positive control), GL and GH: low- and high-dose LB-GABA (100 and 200 mg/kg, respectively). Values are means ± standard deviation (SD); * *p* < 0.05, ** *p* < 0.01, and *** *p* < 0.001, versus normal control according to Tukey’s multiple comparison test; ns, not significant.

**Figure 9 ijms-22-03537-f009:**
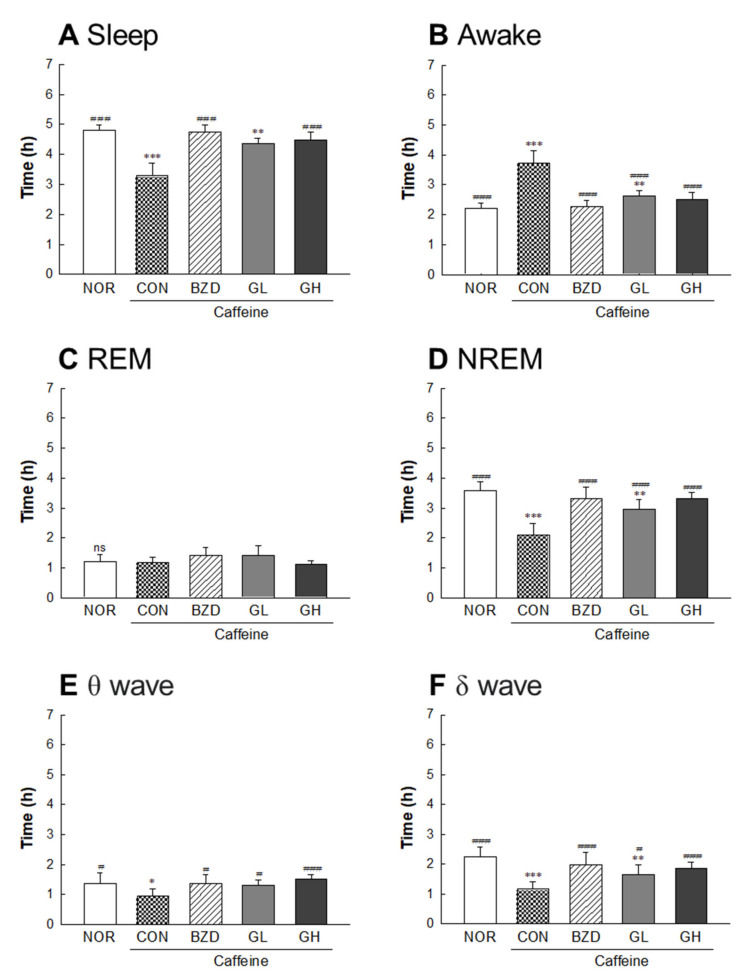
Effects of *Lactobacillus*-fermented γ-aminobutyric acid (LB-GABA) on electrophysiologic pattern in caffeine-induced wakefulness of rats. Caffeine (40 mg/kg) was administered as a stimulant for wakefulness before experiments. GABA was administered orally 1 h before electroencephalography (EEG) analyses, which were conducted for 3 days. (**A**) sleep time, (**B**) awake time, (**C**) duration of REM (rapid eye movement), (**D**) duration of NREM (non-rapid eye movement), (**E**) θ wave of NERM, and (**F**) δ wave of NREM.NOR: normal control (0.9% sodium chloride (NaCl)) group, CON: caffeine only treated (40 mg/kg) group, BZD: benzodiazepine (alprazolam, 300 μg/kg)-treated group (positive control), GL and GH: low- and high-dose LB-GABA (100 and 200 mg/kg, respectively) with 40 mg/kg caffeine treatment. Values are means ± standard deviation (SD); * *p* < 0.05, ** *p* < 0.01, and *** *p* < 0.001 versus normal group; # *p* < 0.05, ## *p* < 0.01, and ### *p* < 0.001 versus caffeine group according to Tukey’s multiple comparison test; ns, not significant.

**Figure 10 ijms-22-03537-f010:**
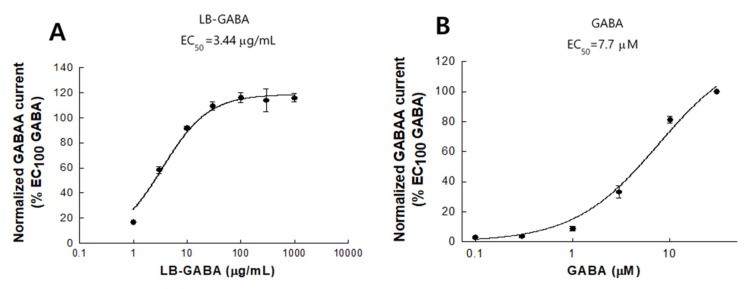
Agonist activity of *Lactobacillus*-fermented γ-aminobutyric acid (LB-GABA) on human GABA_A_ receptor. Agonist activity of LB-GABA on human GABA_A_ (hGABA_A_ α1β2γ2) receptor was tested using IonFlux HT electrophysiological platform. Current response to (**A**) LB-GABA or (**B**) synthetic GABA standard was dose-dependent in HEK293 cell expressing hGABA_A_ α1β2γ2. Experiments were independently performed three times. Maximal effective concentration (EC_100_) of GABA was equal to current level of 30 μM synthetic GABA. EC50, half-maximal effective concentration.

## Data Availability

The data that support the findings of this study are available from the corresponding author upon reasonable request.

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
