# Peer review of "Fermented Gamma Aminobutyric Acid Improves Sleep Behaviors in Fruit Flies and Rodent Models"

_ijms, 2021, doi:10.3390/ijms22073537_

Round 1
Reviewer 1 Report
Very interesting paper. Well conducted research. Excellent English.
Author Response
<Response> Thank you for kind comment
Reviewer 2 Report
Jeong et al presented a manuscript titled: „Fermented gamma aminobutyric acid improves sleep behaviors in fruit flies and rodent models“. The topic is interesting and the manuscript is relatively well written however I have some suggestions and comments:
- There are grammatical errors and typos in the manuscript. Moreover, I noticed that the font is not the same in the whole manuscript. Please thoroughly revise this.
- For the statistical analyses how did you check the normality of distribution? What methods did you use to normalize a variable that was not normally distributed?
- It is not appropriate to use SEM in data presentation. Please change accordingly in all results to SD.
- What P value was set at statistically significant? Add this to statistical methods.
- Some figures are too small and blurry so it is hard to interpret them. Please revise that.
- You are missing information regarding the ethical and animal care committee approval.
- You are missing the limitations in the Discussion section.
- In the Discussion section you shouldn't refer in brackets to your Figures from the Results section (don't put reference to specific figures from the results). Please revise and remove that from the Discussion.
- You should refresh your bibliography with more recent work and you need to style them appropriately for this journal.
- Remove the specific value of EC50 from the Conclusion section.
Reviewer 3 Report
Jeong and coworkers investigated the effect of Lactobacillus breves-fermented gamma-aminobutyric acid (LB-GABA) on sleep behaviors in invertebrate and vertebrate models. They found LB-GABA improved the nighttime sleep behaviors of flies and decreased locomotor activity, which was associated with the increased nighttime sleep duration. The LB-GABA-mediated calming effect was also observed in the DAM system and open-field test using caffeine as a stimulant. In addition, LB-GABA enhanced the sleep-inducing activity of pentobarbital by increasing the sleep onset ratio, shortening the sleep latency, and prolongation of the sleep time. They concluded that LB-GABA effectively improved sleep behaviors in the fruit fly, mouse, and rat models likely via GABAergic actions. The EC50 of LB-GABA on the human GABAA receptor was found to be 3.44 ug/mL. LB-GABA showed a more obvious sleep-promoting effect with an improvement of sleep quality in the caffeine-induced insomnia model. Generally, this study was interesting; materials and methods were well designed; results were convincing; and discussions were reasonable. I have some minor issues with this manuscript. 1. This study seems to be supported by the AMOREPACIFIC Research and Development in Seoul. Therefore, I am worried about the role of the funder. Please certain the role of the funder in this study and manuscript. Furthermore, please remove redundant information that was not proved by this study for reducing the commercial promotion risks. For example, "Moreover, LB-GABA may be a valuable potential non-prescription sleep aid without side effects." "The safety of GABA as a supplement depends on the source [42]. Synthetic GABA has the potential to contain harmful byproducts generated during production [38], ... natural GABA as > 5000 mg/kg in rats. This indicated that LB-GABA is a generally recognized as safe (GRAS) grade." "This study provides evidence to support the potential usefulness of fermented GABA as a readily available safe, food-grade agent for the control of sleep disorders." Otherwise, the readers will be interested in conflicts of interest. 2. "BDZ" is not a formal abbreviation of benzodiazepine. 3. Nonpharmacological treatments were not "recently" suggested as alternatives with milder side effects. Many nonpharmacological treatments such as autogenic training, behavioral treatment, etc, have been suggested for more than 30 years. 4. Please provide a paragraph of study limitations for balancing your positive findings and potential benefits.Author Response
Please see the attachment.

Round 2
Reviewer 2 Report
All concerns and comments are adequately addressed.